# Retinal Proteomic Alterations and Combined Transcriptomic-Proteomic Analysis in the Early Stages of Progression of a Mouse Model of X-Linked Retinoschisis

**DOI:** 10.3390/cells11142150

**Published:** 2022-07-08

**Authors:** Xiuxiu Jin, Xiaoli Zhang, Jingyang Liu, Weiping Wang, Meng Liu, Lin Yang, Guangming Liu, Ruiqi Qiu, Mingzhu Yang, Shun Yao, Bo Lei

**Affiliations:** 1Henan Eye Institute, Henan Eye Hospital, People’s Hospital of Zhengzhou University, Henan Provincial People’s Hospital, Zhengzhou 450003, China; jinxiuxiu@zzu.edu.cn (X.J.); liujingyang@zzu.edu.cn (J.L.); wwp911026@zzu.edu.cn (W.W.); liuguangminghxl@zzu.edu.cn (G.L.); qiuruiqi0828@zzu.edu.cn (R.Q.); phyllis_yang@zzu.edu.cn (M.Y.); jfw@zzu.edu.cn (S.Y.); 2Branch of National Clinical Research Center for Ocular Disease, Henan Provincial People’s Hospital, Zhengzhou 450003, China; 3Academy of Medical Sciences, Zhengzhou University, Zhengzhou 450001, China; ms139@zzu.edu.cn (X.Z.); qingge@zzu.edu.cn (M.L.); liyadyx@zzu.edu.cn (L.Y.)

**Keywords:** X-linked retinoschisis, proteome, RS1, retina, photoreceptor

## Abstract

X-linked retinoschisis (XLRS) is among the most commonly inherited degenerative retinopathies. XLRS is caused by functional impairment of RS1. However, the molecular mechanisms underlying RS1 malfunction remain largely uncharacterized. Here, we performed a data-independent acquisition-mass spectrometry-based proteomic analysis in RS1-null mouse retina with different postal days (Ps), including the onset (P15) and early progression stage (P56). Gene set enrichment analysis showed that type I interferon-mediated signaling was upregulated and photoreceptor proteins responsible for detection of light stimuli were downregulated at P15. Positive regulation of Tor signaling was downregulated and nuclear transcribed mRNA catabolic process nonsense-mediated decay was upregulated at P56. Moreover, the differentially expressed proteins at P15 were enriched in metabolism of RNA and RNA destabilization. A broader subcellular localization distribution and enriched proteins in visual perception and phototransduction were evident at P56. Combined transcriptomic-proteomic analysis revealed that functional impairments, including detection of visible light, visual perception, and visual phototransduction, occurred at P21 and continued until P56. Our work provides insights into the molecular mechanisms underlying the onset and progression of an XLRS mouse model during the early stages, thus enhancing the understanding of the mechanism of XLRS.

## 1. Introduction

XLRS is the main cause of hereditary juvenile macular degeneration in males, with an estimated prevalence ranging from 1/15,000 to 1/25,000 [1]. The hallmarks of the disease include schisis (splitting) of the inner retinal layers, the formation of macular microcysts, and a negative electroretinogram (ERG) response [2,3]. The onset of XLRS is a gradual process, which usually occurs within the first decade of life [4]. Even though visual acuity may remain relatively stable in XLRS boys for years, an acceleration of vision loss due to central retinal atrophy in the fourth to fifth decade of life occurs [4,5]. Previous studies suggest that early interventions may alleviate the progress of XLRS in humans and animals [6,7,8,9]. Nevertheless, the manner in which the malfunction of the RS1 protein induces the degeneration of the inner and outer retinal neurons remains poorly understood.

XLRS is caused by various mutations of the RS1 gene on the X-chromosome [10]. RS1 encodes a secreted protein, retinoschisin (RS1), which functions extracellularly as a covalently linked octameric ring [11,12]. RS1 is secreted by photoreceptors (rods and cones) and bipolar cells, and is distributed mainly in the inner segment (IS) of photoreceptors and the inner nuclear layers (INLs) [13,14]. As an extracellular adhesion protein, secreted RS1 plays crucial roles in determining the structural integrity of the photoreceptor-bipolar synapses and in regulating the fluid balance of retinal cell layers [11,12,15,16]. It is therefore reasonable to consider that retinal schisis observed in XLRS is attributable to dysfunctional RS1 [15,16].

Mouse models that recapitulate typical human XLRS symptoms, including intraretinal schisis and negative pattern ERG, help in delineating the disease pathophysiology [17,18,19,20]. RS1 is apparent in the newly formed outer plexiform layer (OPL) at P7 in mice [21]. RS1-KO retina first shows a schisis lesion in the INLs and disruption of the OPL at P14–18 [17,22,23]. During schisis morphogenesis (P14–18), the outer nuclear layer (ONL) is disorganized but has normal cell counts. Between P24 and 4 months of age, the schisis of the RS1-KO retina diminishes and maintains a stable level. Furthermore, ERG results show greater apparent amplitude reductions of the b-wave than the a-wave. In this period, the outer nuclear layer (ONL) undergoes a slow progressive loss of photoreceptors. Additionally, our previous study also indicates that the thickness of the OS and the IS in RS1-KO mice decreases with age [24,25].

Despite advances in our understanding of the morphological and functional changes during the development of RS1-KO retinas, the underlying molecular changes have not been fully elucidated. Previous studies have noted that RS1 is required for Na^+^/K^+^-ATPase signaling and localization, which show a protective effect against photoreceptor degeneration [26,27]. Weber et al. studied whole genome expression profiling of RS1 KO mouse using DNA-microarrays, and revealed that activation of the MAPK Erk1/2 pathway occurs as early as P7 [28]. By using RNA sequencing, Sieving et al. investigated the transcriptomic changes in RS1-KO mice at P12 and P21. Only a few alterations were observed at P12, whereas 358 genes were differentially expressed at P21 [6]. In contrast to the transcriptomics research, deep proteomic analysis of RS1-KO retinas has not yet been conducted. It has been shown that there are major differences between the two regulatory layers, namely, transcriptome and proteome layers. In addition, the proteomic layer may more accurately reflect the molecular changes in cellular function. Here, we performed proteomic profiling of RS1-KO retinas during the early stages of XLRS progression, including the onset (P15, schisis morphogenesis) and early progression (P56, schisis collapse) stages. Furthermore, we compared our proteomic results with published transcriptomic data, and revealed the relationships between different postal days. This work provides insights into the molecular mechanisms corresponding to the morphological and functional alterations of retinas during the early stages of XLRS progression.

## 2. Results

### 2.1. Morphological and Functional Evaluation of RS1-KO Retinas in Development

XLRS and age-matched litter-mate WT mice were generated in our laboratory. Construction and verification of the XLRS mouse model were detailed in our previous work [25] To evaluate the morphology of RS1-KO retinas in the early postnatal periods, we performed hematoxylin–eosin (HE) staining. In Figure 1a, large cavities within the INL were observed at P15, which became collapsed and shrunken by P56. The nuclear arrangement of the ONL was disordered in RS1-KO retinas at P15 and P56. At P15, the cell numbers in the ONL of RS1-KO retinas were comparable to those in the wild type (WT), whereas a marked decrease was observed at P56. This was consistent with our previous study, which showed that the thickness of the ONL and OS+IS in RS1-KO retinas declined during a long-term observation [24,25].

Next, the retinal functions at different postal days were assessed with dark-adapted ERGs. The a-wave originates mostly from photoreceptors, and the b-wave mostly represents the activities of the post-receptoral bipolar [29,30,31,32]. In Figure 1b, compared with the age-matched WT mice, the RS1-KO retinas showed a normal or nearly normal a-wave, but a decreased b-wave at P15. Both a- and b-wave amplitude decreased at P56. The functions of the photoreceptors were relatively preserved at P15, whereas they declined at P56. The functions of bipolar cells or neurotransmitter release were impaired since P15.

### 2.2. Proteomic Features of Retina in XLRS Mice

To explore the molecular mechanisms underlying the structural and functional alterations in RS1-KO retinas with the progression of XLRS, we performed proteomic analysis. The experimental strategy is shown in Figure 2a. The retinas were separated into four groups based on their age (P15 or P56) and genotype (RS1-KO or age-matched WT). Whole proteins of the retinas in the groups (*n* = 4 mice per group) were analyzed by liquid chromatography–mass spectrometry (LC–MS) using label-free quantification via a DIA approach. A total of 4779 proteins were quantified in the RS1-KO and the age-matched WT retinas at P15. A total of 4201 proteins were quantitatively detected at P56; among these, 3632 proteins were common to P15 (Figure 2b, Appendix A).

Gross protein differences between samples were visualized by principal component analysis (PCA) using the normalized log2 intensity values of the specific proteins. Apparently, the samples were segregated well by age P15 vs. P56 (PC1, 29.79% variance) and the genotype RS1-KO vs. WT (PC2, 18.38% variance) (Figure 2c–e). Additionally, P15_RS1-KO/P15_WT clustered closer than P56_RS1-KO/P56_WT (Figure 2d,e), indicating that age had a significantly greater impact than genotype, and proteomes of RS1-KO and WT retinas were more similar at P15 than those at P56. Next, to make the results clearer, we show heatmaps of DEPs at P15 and P56 in Figure 2f. Apparently, red (high intensity) and blue (low intensity), which represent specific protein abundance, were better separated in samples at P15 and P56. Moreover, samples were clustered hierarchically, as labelled on the top of the heatmap. Columns are grouped by their genotype, indicating that samples were segregated well by genotype RS1-KO vs. WT at P15 and P56.

As shown in Figure 2f,g, there were 251 significantly changed proteins (*p*-value < 0.05) at P15, of which the abundance of 211 proteins changed by 1.5-fold (accounting for 84.06%), including 108 upregulated and 103 downregulated proteins (Figure 2h,f). Additionally, 475 proteins were significantly changed (*p*-value < 0.05) at P56 (Figure 2g), but only 165 of them changed by at least 1.5-fold (accounting for 34.74%) (Figure 2g,f), including 85 upregulated and 80 downregulated proteins (Figure 2i). Among them, 31 significantly changed proteins (*p*-value < 0.05) were shared by P15 and P56, and all of them were changed by 1.5-fold (Figure 2g). Therefore, there were fewer significantly changed proteins (*p*-value < 0.05) in the P15_RS1-KO retinas, but the magnitude of these changes was relatively larger. We speculated that in the early developmental stage of RS1-KO retinas, those proteins that were differentially expressed may be the “primary” pathological mechanism-related proteins, whereas those at P56 could be the “secondary” pathological mechanism-related proteins, of which there were more in number but having smaller fold changes.

### 2.3. Differentially Expressed Gene Sets in XLRS Onset

Conventional enrichment analysis focuses on comparing the DEPs between two groups, relying on a few proteins that are significantly upregulated or downregulated. This method may miss some key information due to unreasonable screening parameters. To comprehensively present RS1-related functional alternations occurring in the retina as XLRS progresses, GSEA (v4.1.0, downloaded from http://www.broadinstitute.org/gsea, accessed on 5 July 2022) was performed. GSEA evaluates gene expression on the level of gene sets that are based on prior biological knowledge, allowing a more interpretable and reproducible analysis of the gene expression data [33]. GSEA directly analyzes the expression levels of all proteins, which can detect gene sets with insignificant but consistent differential expression trends. As no gene selection step (*p*-value or fold change cut-off) is used, analysis using the GSEA approach is unbiased [33,34,35]. The enrichment results from GSEA, focusing on the gene oncology (GO) biological process, are presented in Figure 3a,b.

At P15, the top five downregulated gene sets included the positive regulation of peptide hormone secretion, phototransduction, detection of light stimulus, phototransduction of visible light, and the positive regulation of hormone secretion. Swelling-induced peptide secretion represents an important cellular reaction whereby material stored in the secretory vesicles is expulsed as a secretory burst of peptide hormones or enzymes from cells such as neurons, endocrines, or leukocytes [35]. Cell shrinkage often has a suppressive effect on this type of secretion [36,37]. Therefore, the downregulation of the promotion of (peptide) hormone secretion indicated that cells in the RS1-KO retinas can be shrunk by external stimulation at P15. Figure 3c shows there were 29 proteins involved in positive regulation of peptide hormone secretion, 15 of which were the core downregulated proteins, including Psmd9, Prkce, Pfkfb2, Pck2, Mcu, Lrp1, Ppp3cb, Rac1, Fga, Aimp1, Acsl4, Nlgn2, Cltrn, Aacs, and Fgg. Moreover, the downregulation of phototransduction and the detection of light stimulus indicated that the functions of proteins related to photoreceptors and bipolar cells were altered at P15. Combined with the ERG results at P15, we inferred that those proteins related to photoreceptors and bipolar cells underwent significant changes; however, the bipolar cells showed an apparent impairment in function. Specifically, the enriched phototransduction contains six core downregulated proteins, including Opn1sw, Rho, Gucy2f, Rcvrn, Plekhb1, and RS1, which may be related to the impaired functions of bipolar cells, or represent a subsequent functional decline of photoreceptors.

The upregulated gene sets included the regulation of the type I interferon (IFN-I)-mediated signaling pathway, response to IFN-I, nucleobase metabolic process, response to nerve growth factor, and non-motile cilium assembly. In Figure 3e, regulation of INF-I-mediated signaling pathway contains six core upregulated proteins, namely, Irf3, Mul1, Mavs, Abce, Cactin, and Ptpn2. IFN-I controls both acute and chronic viral infections [38,39].

### 2.4. Differentially Expressed Gene Sets in XLRS Early Progression

GSEA results for the RS1-KO retinas at P56 are shown in Figure 3b. Gene sets including the positive regulation of Tor signaling, neural retina development, spermatid differentiation, protein homo-oligomerization, and the regulation of megakaryocyte differentiation were downregulated in the RS1-KO retinas, whereas NTMCP nonsense-mediated decay, tube formation, midbrain development, cell recognition, and the response to amyloid beta were upregulated. Tor and its mammalian ortholog mTOR are serine–threonine kinases that respond to growth factors and promote appropriate changes in cell proliferation and survival [40,41]. Therefore, alteration of vitality-related functions, such as neural retina development, spermatid differentiation, and regulation of megakaryocyte differentiation in RS1-KO retinas at P56 may be related to the Tor pathway. Specifically, as shown in Figure 3f, the pathway of positive regulation of Tor signaling contains 21 proteins, and core downregulated proteins are Lamtor5, Rictor, Hdac3, and Sik3. As indicated by Figure 3g, 25 proteins are involved in neural retina development, and Bbs4, Atp8a2, Slc1a1, Slc17a7, Prom1, Rom1, Sdk1, and RS1 are the core downregulated proteins. Moreover, the core enriched proteins involved in NTMCP nonsense-mediated decay are PARN, RPS23, RPL8, RPL35, etc. (Figure 3h). NTMCP nonsense-mediated decay is a conserved cellular quality control mechanism that targets ~10% of unmutated mammalian mRNAs to facilitate appropriate cellular responses (adaptation, differentiation, or death) to environmental changes [42,43,44]. Therefore, the upregulation of NTMCP may contribute to cellular quality control in response to RS1 depletion.

### 2.5. RS1 Deletion Itself Induces the Downregulation of Tor Pathway

To demonstrate whether the Tor pathway is downregulated in XLRS mice, we performed WB validation with canonical phosphorylated proteins in the Tor pathway and analyzed their relative gray values (Figure 4a,b). Apparently, phosphorylation levels of P70s6k, 4ebp1, and mTor were significantly downregulated in XLRS retinas at P56, indicating that the Tor pathway was inhibited in XLRS mice at P56. The retina is a complex structure composed of a variety of cells, and cell numbers in the ONL of RS1-KO retinas were decreased at P56. This raises the question of whether the downregulated Tor signaling pathway is a secondary effect due to structural changes (e.g., ONL cell reduction), or whether RS1 deletion itself induces the signaling pathway alternation (i.e., a primary effect). To address this question, we constructed the mouse embryonic fibroblast (MEF) cell line in XLRS mice, and investigated the changes in the Tor signaling pathway in this cell line. The purity of MEF cell lines was validated by immunofluorescence using Vimentin, an antibody specific for MEF cells (Figure 4c). As shown in WB analysis (Figure 4d,e), RS1 was deleted in XLRS mouse-derived MEF cells. Compared with WT mouse-derived MEF cells, phosphorylation levels of P70s6k, 4ebp1, and mTor were also significantly downregulated in XLRS mouse-derived MEF cells. The above results indicate that RS1 deletion itself induces the downregulation of the Tor pathway.

### 2.6. Same and Contrasting DEPs at the Onset and Early Progression of XLRS

Differences in protein expression are the driving force behind changes in biological functions and phenotypes. In Figure 2a, 31 DEPs (*p*-value < 0.05, 1.5-fold change) were shared by P15 and P56. However, only 15 displayed the same regulation trends, including 7 downregulated and 8 upregulated proteins. In Figure 5a, Cdc42, Cdkal1, Ctsf, Emc4, Etf1, Smyd5, Tubb4a, and Vapa were upregulated, but Dguok, Hdgf, Kcnb1, Mcrip1, Ndufab1, RS1, and Trafd1 were downregulated at both P15 and P56. Specifically, Cdc42 and Tubb4a belong to the Rho GTPases, which play crucial roles in triggering multiple immune functions. The data were consistent with the dysregulation of immune response at an early age in RS1-KO retinas according to the transcriptional analysis [6,45]. Interestingly, 16 proteins showed contrasting expression trends at P15 compared with P56. As listed in Figure 5b, nine proteins, namely, Avl9, Gart, Scamp5, Scyl2, Sipa1l3, Tsr2, Ttc8, Ube2m, and Znf 787 were upregulated at P15, whereas they were downregulated at P56. Specifically, Avl9 has functions in cell migration, and upregulation of Avl9 may be a mechanism underlying nuclei migration in the retina [46,47]. In addition, seven proteins, namely, Chl1, Crip2, Eif3i, Impg1, Impg2, Pon1, and Sec22b, were downregulated at P15 but upregulated at P56. Notably, Chl1 is a neural cell adhesion molecule L1-like protein that acts similarly to RS1 as a cell adhesion protein [48]. Moreover, Impg1 and Impg2 are interphotoreceptor matrix proteoglycan proteins that play an important role in photoreceptor growth [49,50,51]. Therefore, in the early stage of XLRS progression, RS1-KO retinas may exhibit autoregulatory responses in cell adhesion and photoreceptor growth.

### 2.7. Functional Enrichment Analysis of the DEPs in XLRS Progression

To further explore the functions of the DEPs (1.5-fold change, *p* < 0.05), we performed pathway and GO enrichment analysis using Metascape (http://metascape.org/, accessed on 11 November 2021). Enrichment results with FDR < 0.25 are shown in Figure 5c,d (Appendix A). At P15, the DEPs were mainly involved in the metabolism or location of macromolecules and development-related pathways, including the metabolism of RNA, RNA destabilization, camera-type eye morphogenesis, regulation of protein-containing complex assembly, protein localization to membrane, protein-containing complex localization, and cellular response to chemical stress. By comparison, at P56, the DEPs were mostly enriched in visual functions, such as visual perception, visual phototransduction, and photoreceptor cell differentiation. Specifically, the differentially expressed proteins, including Cntf, Fgf2, and Slc4a7, were enriched in retinal cell programmed cell death, which is consistent with the decline of ONL and OS+IS thickness in the progress of XLRS [24,25].

### 2.8. Changes in Subcellular Location in XLRS Progression

Retina is a multi-laminated structure that contains functionally and structurally diverse neurons, with highly specialized-cellular and compartmental sub-cellular organization. Moreover, RS1 is a secreted protein mainly produced by bipolar cells, and is involved in the composition of the INL (nucleus of BP), IPL, and OPL (mainly cytoplasm and some organelles of BP). To explore the relationships between the progression of the disease and the distribution of RS1-altered proteins, we performed GO subcellular localization analysis of the DEPs. Figure 6a,b shows that the distribution of DEPs at P15 is relatively simple, mainly concentrated in the cytoplasm, nucleus-related regions (including the nucleus, nucleoplasm, and nuclear matrix), extracellular parts (extracellular exosome), etc. By comparison, the DEPs appear to be more widely distributed at P56, concentrated in the cytoplasm, nucleus-related regions (including nuclear envelope, perinuclear region of cytoplasm), extracellular parts (extracellular exosome and extracellular space), membrane, photoreceptor outer and inner segments, etc., involving almost all layers of the retina. Moreover, the number of DEPs in the nucleus decreased, and those in the membrane, photoreceptor outer segment, and photoreceptor inner segment increased significantly in early-stage progression of XLRS. The alternations in proteins in the cytoplasm, nucleus-related regions, and extracellular exosome at P15 echo the splitting of the INL and disorganization of the ONL with the onset of XLRS. The increased tendency of DEPs in the membrane, and photoreceptor outer and inner segments, at P56, coincides with the functional impairment of the post-receptoral bipolars. The wider distribution of DEPs at P56 may be associated with the increased secondary response proteins of RS1. Therefore, we conjectured that, compared to P15, the RS1 secondary response proteins may increase at P56, and may be concentrated in the membrane, extracellular parts, photoreceptor outer segment, or photoreceptor inner segment.

### 2.9. Correlation between the Proteome and Published Transcriptome Data in RS1-KO Retinas

To reveal the connections between the transcriptome and the proteome, and to discover important pathways and functional proteins, we performed a combined analysis of proteomic and published transcriptomic data [6]. The correlation analysis revealed that the proteome of RS1-KO retinas at P15 exhibited a modest correlation with the transcriptome at P12 but not with the transcriptome at P21 (Figure 7a,b). By comparison, the altered proteome at P56 was positively correlated with that at P21 (Figure 7c). Apparently, the Pearson correlation coefficients (R) between mutation-dependent changes detected in the transcriptome and proteome are relatively low. To prove that this phenomenon is reasonable and reliable, we analyzed the reported transcriptome data of RS1 KO retinas at P12 and P21. Results showed that the R-value between the transcriptomic layers in RS1-dependent changes detected at P12 and P21 was 0.06, comparable with our results (Appendix A). The lower correlation indicates that the expression of mouse retinal proteins varies greatly at different postal days in early stages. Moreover, considering the fact that protein abundance is regulated at multiple post-transcriptional levels, the correlation results in our work are reasonable and reliable. Furthermore, the moderate positive relationship (R = 0.32) at P21 vs. P56 indicates that significant pathological changes may occur between P15 and P21, and some pathological may change continue up to P56.

A total of 33 proteins showed the same changes when comparing the data for the transcriptome at P21 with those of the proteome at P56, including 18 upregulated and 15 downregulated proteins (Figure 7d). Expression levels of mRNAs of Abca4, Rdh12, Reep6, and Room1 were validated by qPCR. Results confirmed that Reep6, Rom1, Rdh12, and Abca4 were significantly downregulated in RS1 KO retinas (P56) (Figure 7e). Next, we explored the functions of these 33 proteins using Cytoscape GoClue-based enrichment analysis. As shown in Figure 7f, the downregulated proteins including Unc119, Opn1mw, Abca4, Rdh12, Reep6, Gnb1, Rom1, Arr3, RS1, and Rgs9bp were found. These proteins are involved in the detection of visible light or visual perception biological processes. Proteins including Opn1mw, Arr3, Nxnl1, Rom1, and Abca4 may mainly affect functions of the photoreceptor outer segment or the photoreceptor outer segment membrane (Figure 7g). KEGG pathway analysis indicated that visual phototransduction enriched by Opn1mw, Gnb1, Rgs9bp, Apoe, Rdh12, and Abca4 may be downregulated in RS1-KO retinas (Figure 7h). Therefore, functional impairments, including in detection of visible light, visual perception, and visual phototransduction, occurred at P21 and continued up to P56. These findings are in agreement with the retinal function and morphological changes observed in RS1-KO mice.

## 3. Discussion

Functional impairment of RS1 is the leading cause of XLRS. The onset of XLRS is a gradual process, with typical structural and functional alterations. Previous studies have suggested that early intervention may alleviate the progress of XLRS in humans and animals. Therefore, elucidating the molecular mechanism of early-stage XLRS is of great significance. We reported the proteome alterations in the retina at an early stage of a mouse model of XLRS, including the onset (P15) and early progression (P56), and revealed the underlying mechanisms corresponding to morphological and functional alterations.

Extensive research has shown that RS1 plays an important role in the adhesion of bipolar cells, which is a major cause of schisis (splitting) of the INL in XLRS. Disorganization of the INL disrupts the transmission of synaptic signals from photoreceptors to ON-bipolar cells, leading to a progressive loss of vision [52,53]. In addition to bipolar cells, RS1 is also primarily localized in the IS of photoreceptor cells [52]. Photoreceptors, including rods and cones, form the outermost half of the retina. They produce electrical responses upon capturing photons. In this work, as indicated by GSEA results, photoreceptor proteins in the detection of light stimulus may be damaged at an early stage in XLRS retinas at P15. Differentially expressed proteins of XLRS at P56 also showed enrichment results related with photoreceptors, such as visual perception, photoreceptor cell differentiation, photoreceptor cell maintenance, and the canonical retinoid cycle in rods (twilight vision). Moreover, the transcriptomic and proteomic data revealed decreased levels of proteins in photoreceptors, including Rdh12, Reep6, Rom1, Opn1mw, and Arr3. Our previous report also revealed that the thickness of OS and IS in XLRS mice and patients decreased with age, and reduction in a-wave amplitude correlated well with the decline in ONL and OS+IS thickness [24,25]. Based on these findings, we infer that photoreceptor cells underwent progressive dysfunction from the early stages.

The number of photoreceptors decreased as XLRS progressed, which can lead to progressive vision loss. Cell death is a key regulator of the number of neurons in the nervous system during development [54,55,56,57]. A controlled balance between proliferation and death is a known process for adjusting the size and the cytoarchitecture of neuronal structures. As indicated by the enrichment of the differentially expressed proteins, biological processes such as retinal cell programmed cell death, photoreceptor cell differentiation, and photoreceptor cell maintenance were altered at P56. Therefore, RS1 dysfunction may affect the balance between proliferation and death of cells in the retina. As indicated by GSEA results, the underlying mechanisms of RS1-related proliferation control may be associated with pathways such as the positive regulation of Tor signaling and the NTMCP nonsense-mediated decay in RS1-KO retinas. Moreover, it should be considered that a decrease in the number of photoreceptors will inevitably lead to a decrease in proteins in photoreceptors. Studies on whether the downregulated photoreceptor-related proteins are directly regulated by RS1 dysfunction, or by secondary responses caused by an RS1-induced decrease in photoreceptors, are currently on-going in our laboratory.

The formation and sinking of the cleavage cavity are the most important morphological changes in XLRS progression [3,5]. At the onset of XLRS, a large cleavage cavity in the INL was observed. The structure of the ONL was disorganized, with some nuclei displaced into the OPL and the inner segment layers. In our results, several cell shrinking-induced biological processes were downregulated in schisis morphogenesis (onset, P15, Figure 3a), such as the positive regulation of peptide hormone secretion [35]. Moreover, cell migration-related Avl9 was specifically upregulated [46,47], which may be the mechanism underlying ONL disorganization and nuclei migration at the onset of XLRS. In contrast, Avl9 was downregulated at P56, and the corresponding nuclei displacement was alleviated, and had disappeared in long-term observations [17]. Considering the shrinking of schisis at P56, we also found that Chl1, another adhesion-related protein, was specifically upregulated at P56 [48].

Mouse retinas are not fully developed at P15, and the effect of RS1 dysfunction on the retina may have not fully exhibited [58,59]. The ERG results at P15 indicate that bipolar cells were impaired but functions of the photoreceptors were relatively preserved. By comparison, in mature XLRS retinas, the functions of both the bipolar cells and photoreceptors appeared to be impaired. Correspondingly, as indicated by the subcellular location analysis, the distribution of differentially expressed proteins is relatively simple at P15, and mainly concentrated in nuclear-related structures, which echoes the splitting of the INL and disorganization of ONL with the onset of XLRS. By comparison, in mature XLRS retinas (P56), the distribution of differentially expressed proteins became more widespread, involving almost all layers of the retina. The above conclusions verify that the influence of RS1 dysfunction increased as XLRS progressed during early stages, as considered in terms of protein alteration. We believe this indicates that early intervention may be more effective in remission of XLRS.

In summary, we present a global proteomic profile of RS1-KO mice retinas at the onset and during the early progression stages of XLRS. This study provides insights into the molecular mechanisms underlying the onset and progression of XLRS, and presents solid data for future research on XLRS and its early intervention.

## 4. Materials and Methods

### 4.1. Experimental Mice

Retinoschisin knockout (RS1-KO) mice and age-matched litter-mate C57BL/6 WT mice were generated in our laboratory. The RS1 KO mouse was designed with the CRISPR/Cas9 technique by targeting RS1-201 (NM_011302.3). Construction and verification of the XLRS mouse model were detailed in our previous work [25]. Animal experimental procedures were conducted in accordance with the Association for Research in Vision and Ophthalmology (ARVO) Statement on the Use of Animals in Ophthalmic and Vision Research, and were also approved by the Animal Ethics Committee of Henan Eye Hospital/Henan Eye Institute.

### 4.2. Histologic Analysis

Mice were sacrificed by cervical dislocation at P15 or P56. Eyecups for immunohistochemistry were immersed in 4% paraformaldehyde (PFA) overnight. Fixed eyes were dehydrated in ethanol, cleared in xylene, and embedded in paraffin. PFA-fixed eyes used for confocal immunofluorescence microscopy were embedded in low-melting-temperature agarose (Sigma-Aldrich, St. Louis, MO, USA) and sectioned. Paraffin sections were then stained with hematoxylin–eosin (HE).

### 4.3. ERG Assessment

ERGs were recorded according to a previously described method [60,61]. Briefly, ERGs of the anesthetized mice were recorded after the overnight dark adaptation. A quantity of 4% chloralhydrate solution was used for the anesthesia of mice. Oxybuprocaine hydrochloride eye drops (Benoxil^®^ 0.4%) were used as an ophthalmic topical anesthetic. Pupils were dilated with tropicamide before recording. Needle electrodes were inserted into the back and the tail as reference and ground leads. All procedures were performed under dim red light. Full-field ERGs were recorded with the RetiMINER-C visual electrophysiology system (AiErXi Medical Equipment Co., Ltd., Chongqing, China). For the dark-adapted ERGs, stimulus intensities ranging from −3 to 1 log cd-s/m^2^ were used. After light adaptation, light-adapted ERGs were recorded using strobe-flash stimuli (0 and 1 log cd-s/m^2^) superimposed on the background light.

### 4.4. Generation of MEF Cells

C57B16 WT (or RS1 KO) embryos were harvested on gestational day 13.5. Heads and visceral tissue were removed. Remaining tissue was disassociated with trypsin and physical disruption and plated in complete MEF media (high glucose [H-21] DMEM, 10% FBS, non-essential amino acids, L-glutamine, Penn/Strep, 55 μM beta-mercaptoethanol) (P0), and then cultured at 37 °C in a 5% CO_2_ incubator. MEFs were expanded to P3 for use.

### 4.5. Real-Time PCR

Total RNA from WT and RS1 KO (P56) mouse retinas were extracted using TRIzol reagent (Thermo Fisher Scientific, Waltham, MA, USA). Complementary DNA (cDNA) was generated using the PrimeScript^®^ RT reagent kit (Takara Biotechnology, Dalian, China). Real-time PCR was performed according to the manufacturer’s instructions with the ABI Prism 7500 system (Applied Biosystems, Thermo Fisher Scientific). The oligonucleotide primer sequences were as follows: Rdh12_Forward, GATACTGCAGTGCTTTTGCCTATGG and reverse, GAGCCGCCATAGCAAACACAGCAGG; Reep6_Forward, AGCGGAAGAGCATTGGACCTA and reverse, GTGGGGTCTGACTTTACTTGG; Rom1_Forward, CTCCAACCCCGTATCCGTTTG and reverse, GAGCAGGGAATGAACAAGAGG; Abca4_Forward, TATGCCAGCCTTTTCCGAGAGC and reverse, TCCAGCATCCTCTGTGACCTTC. Expressions of mRNA were normalized to the endogenous reference gene β-actin, and all samples were detected in triplicate.

### 4.6. Sample Preparation for Proteomic Analysis

Eyes were obtained from mice with ages at P15 (or P56), then retinas were isolated from the eyes. Four replicates of retinas/per group were used for proteomic analyses. Retinas were lysed in 8 M urea lysis buffer (50 mM NH_4_HCO_3_, 8 M Urea, 1 mM NaF, 1 mM Na_3_VO_4_), supplemented with protease inhibitor cocktail, followed by 2 min ultrasonic sonication on ice. Protein lysates were centrifuged (14,000× *g*, 10 min) and the supernatants were collected and quantified using BCA (Solarbio, Beijing China). A quantity of 100 μg of proteins was reduced by 10 mM DTT (Sigma-Aldrich, St. Louis, MO, USA) at 37 °C for 1 h, and alkylated by 40 mM iodoacetamide (IAA, Sigma-Aldrich, St. Louis, MO, USA) in the dark at room temperature for 1 h. Samples were subsequently centrifuged at 14,000× *g* for 40 min at 15 °C in pre-equilibrated ultrafiltration tubes (10 kDa). Then, tubes were washed 3 times using 50 mM NH_4_HCO_3_ at 14,000× *g*. Next, pretreated proteins were digested with trypsin (1:50 enzyme to protein) overnight at 37 °C. Tryptic peptides were then centrifuged at 14,000× *g* for 20 min and the peptides were collected in two subsequent elution steps by H_2_O_2_. Then, 0.1% formic acid (FA) was added to stop digestion. Next, the digested peptides were vacuum-dried at 60 °C. Digested peptides were dissolved in 0.1% FA and quantified using a Nanodrop 2000 spectrophotometer (Thermo Scientific, Waltham, MA, USA) for LC–MS/MS analysis.

### 4.7. LC–MS/MS Analysis

LC–MS/MS was performed in an Eksigent NanoLC 415 (AB Sciex, Framingham, MA, USA) connected to a Triple TOF 6600 mass spectrometer (AB Sciex, Framingham, MA, USA). After injection, samples were loaded into a trapping column (100 µm × 2 cm; C18: 3 µm, 120 A) by mobile phase A (0.1% FA, 2% acetonitrile). Then, peptides were separated on an in-house fabricated column (150 μm × 15 cm × 1.9 μm; C18: 1.9 μm, 120 A) with a gradient extending from 5% to 35% mobile phase B (0.1% FA, 97.9% acetonitrile, 2% water, *v*/*v*/*v*) for 91 min at a flow rate of 400 nL/min. Next, the separated peptides were detected by MS.

In MS analysis, data-dependent acquisition mode was used to build a spectra library for protein identification and quantification by DIA. For DDA, the accumulation times for TOF-MS and MS/MS scanning were set at 0.25 and 0.04 s, respectively. The other parameters were as follows: ion charges, +2 to +5; mass scan range, 300–1500 *m*/*z*; ion exclusion time, 16 s; mass tolerance, 50 ppm; candidate ions per cycle, 60. In DIA analysis, a variable window assay calculator (AB Sciex, version 1.1, Framingham, MA, USA) was used to optimize the 60 scanning windows. For MS1, the accumulation time was set as 50 ms, and as 40 ms for MS_2_. The ion spray floating voltage was 2300 V.

### 4.8. LC–MS/MS Data Processing

DDA data were processed in Proteinpilot software (5.0.1). For data processing, the parameters were as follows: species: “mice”; sample type: “identification”; special factor: “urea denaturation”; enzyme digestion method: “trypsin”; cysteine: “alkylated with iodoacetamide”; unique peptide score for the false discovery rate analysis (upper limit): “0.05”. The UniProt Swiss mouse database (UniProt release 2019_07) was used as a library database for DIA. For DIA data processing, PeakView software (Version 2.2, Framingham, MA, USA) with the SWATH 2.0 plug-in was used. We used 6 peptides with 6 transitions per peptide for protein quantification. The confidence threshold was set at 99%, and the FDR was 1%. Modified peptides were excluded. The allowed mass deviation was 20 ppm, and the retention time deviation was 6 min. One to two peptides with high spectral quality per 10 min were secreted for retention time correction. Mass spectra were searched using ProteinPilot software (v.5.0.1, AB Sciex, Framingham, MA, USA) using the UniProt Swiss mouse database (v.2019_07).

### 4.9. Bioinformatic and Statistical Analysis

Protein intensities were transformed by log2, and then normalized using the R package (PreprocessCore/normalize.median). P15_WT (or P56_WT) median abundance was set as the reference in analysis, and protein abundances of other samples were median centered accordingly. Intensities of proteins were normalized based on the total intensity of each sample. Log2 (FoldChange) was also calculated. P-values between samples were calculated by *t*-test. Plots for PCA analysis were generated by ggplot. Volcano plots were generated by “ggplot2” in R. Heatmaps using the “pheatmap” package. GSEA was performed with GSEA desktop software (v4.1.0) using the GO BP (c5.go.bp.v7.4.symbols.gmt) database. The published RNA-Seq data were downloaded as previously reported [6]. Pathway and GO enrichment analysis were performed using Metascape (http://metascape.org/, accessed on 11 November 2021).

## Figures and Tables

**Figure 1 cells-11-02150-f001:**
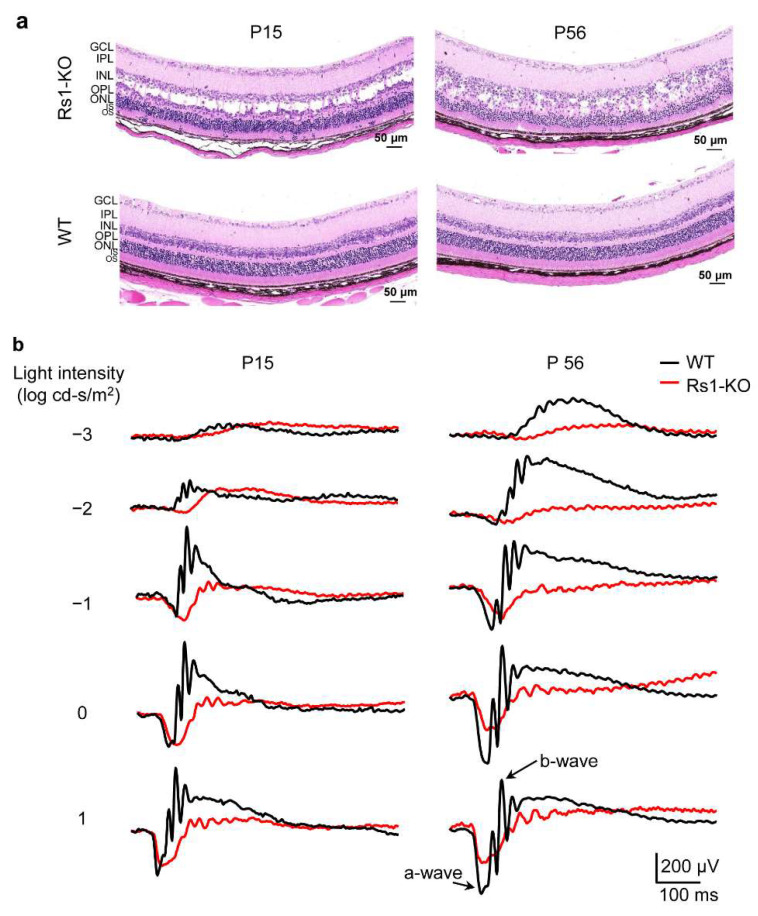
Morphological and functional evaluation of RS1-KO retina. (**a**) RS1-KO and age-matched WT mouse retina sections with HE staining. (**b**) Conventional ERG amplitudes elicited with a series of light intensities in RS1-KO mice and age-matched WT mice. P, postal day; KO, knock out; WT, wild type; HE, hematoxylin–eosin; ERG, electroretinogram; GCL, ganglion cell layer; IPL, inner plexiform layer; INL, inner nuclear layer; ONL, outer nuclear layer; OPL, outer plexiform layer; OS, outer segment; IS, inner segment.

**Figure 2 cells-11-02150-f002:**
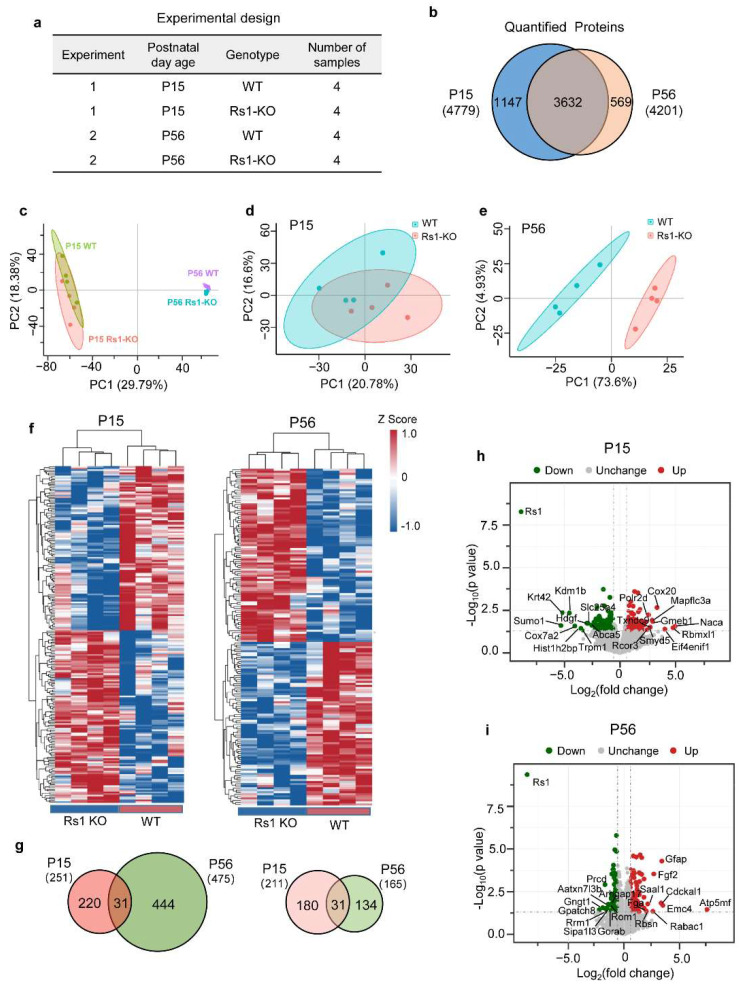
Proteome overview of RS1-KO retinas. (**a**) The experimental strategy. Four groups of retinas were separated into two experiments based on their age (P15 or P56) and genotype (RS1-KO or WT) (*n* = 4 mice per group). (**b**) Venn diagram showing the overlap of quantified proteins in retinas at P15 (blue) and P56 (orange). (**c**–**e**) PCA of the gross protein differences between samples (**c**), and between samples with different genotypes at P15 (**d**) and P56 (**e**). (**f**) Clustered heatmaps of DEPs in RS1 KO and WT retinas at P15 (left) and P56 (right). Samples were clustered via calculating the Euclidean distance between centroids. (**g**) Venn diagram showing overlap of the significantly changed proteins with *p*-value < 0.05 (left) and DEPs with *p*-value < 0.05 and fold change > 1.5 (right) in retinas at P15 (blue) and P56 (orange). (**h**) Volcano plot showing 108 upregulated (red) and 103 downregulated (green) proteins in RS1-KO retinas at P15 (fold change > 1.5, and *p*-value < 0.05). (**i**) Volcano plot showing 85 upregulated (red) and 80 downregulated (green) proteins in RS1-KO retinas at P56 (fold change > 1.5, and *p*-value < 0.05). PCA, principal component analysis; DEP, differentially expressed protein.

**Figure 3 cells-11-02150-f003:**
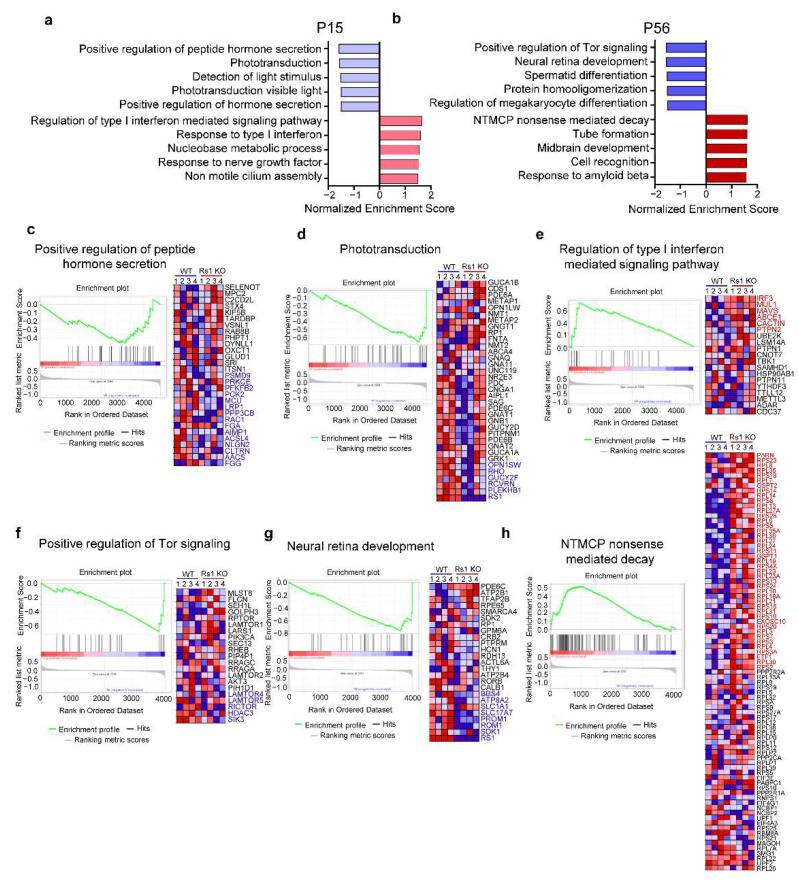
GSEA analysis of RS1-KO retinas. (**a**,**b**) Enriched gene sets of RS1-KO retinas at P15 (**a**) and P56 (**b**) using GSEA (version_4.1.0; database_c5.go.bp.v7.4.symbols.gmt). Top five upregulated (red) and downregulated (blue) gene sets are shown. (**c**–**e**). Enrichment plot (left) and heatmap (right) of the enriched gene sets at P15, including the positive regulation of peptide hormone secretion (**c**), phototransduction (**d**), and regulation of type I interferon-mediated signaling pathway (**e**). (**f**–**h**) Enrichment plot (left) and heatmap (right) of the enriched gene sets at P56, including the positive regulation of Tor signaling (**f**), neural retina development (**g**), and NTMCP nonsense-mediated decay (**h**). Proteins involved in the enriched gene sets are labeled on the right of the heat map, and core regulated proteins in specific gene sets are labeled with red (upregulated) or blue (downregulated) according to the direction of change. The enrichment score (ES) value calculation process is shown on the upper part of the enrichment plots. The particularly pronounced peak represents the ES value of the enriched gene set. Each line in the middle of the enrichment plots represents a gene in the gene sets and its ranked position in the gene list. Red represents positive correlation with phenotype (high expression), blue means negatively correlated with phenotype (low expression). GSEA, gene set enrichment analysis; NTMCP, nuclear transcribed mRNA catabolic process; ES, enrichment score.

**Figure 4 cells-11-02150-f004:**
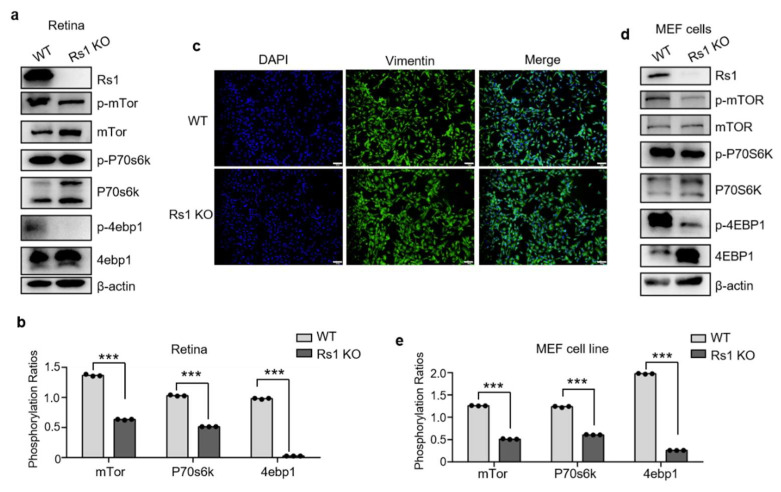
Validation of the Tor signaling pathway. (**a**) WB analysis of XLRS retinas at P56. (**b**) Statistical histogram of relative gray value of proteins in (**a**). (**c**) Immunofluorescence images of MEF cells. DAPI fluorescence (blue) marking the cell nuclei. Vimentin (green), an antibody specific for MEF cells, were used to validate the purity of MEF cell lines. Bar represents 100 µm. (**d**) WB analysis of MEF cells derived by WT (left) or RS1 KO (right) mouse. (**e**) Statistical histogram of relative gray value of proteins in (**d**). WB results were quantified using ImageJ software (Version 1.52a, NIH, Bethesda, MD, USA) and are presented as ratios of the grey value (phosphorylation of target protein vs. target protein). Differences between groups were assessed by one-way ANOVA. ***, *p* < 0.001.

**Figure 5 cells-11-02150-f005:**
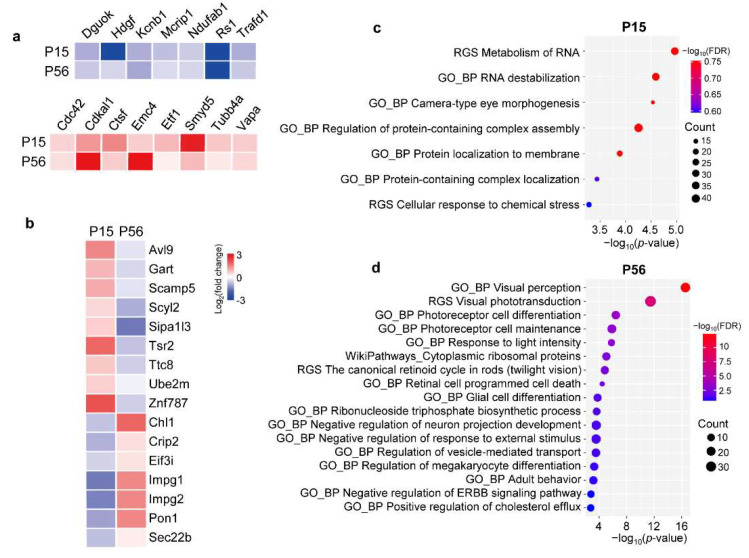
DEPs in RS1-KO retinas and their related functions. (**a**,**b**) Heatmap showing DEPs (*p*-value < 0.05, 1.5-fold change) shared by P15 and P56. Fifteen proteins displayed the same regulated trend, including seven downregulated and eight upregulated proteins (**a**). Sixteen proteins showed contrasting expression trends in P15, compared with P56 (**b**); (**c**,**d**) Metascape enrichment results associated with DEPs in RS1-KO retinas at P15 (**c**) and P56 (**d**) compared to age-matched WT controls. DEP, differentially expressed protein; RGS, reactome gene sets; BP, biological process.

**Figure 6 cells-11-02150-f006:**
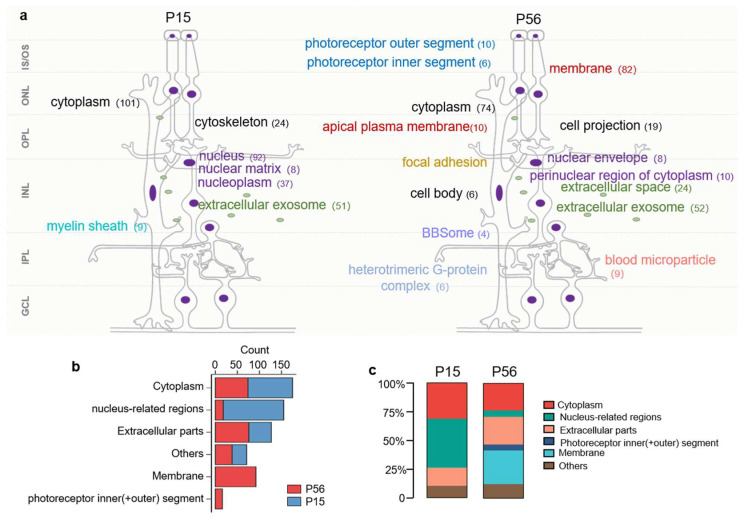
GO subcellular location analysis of the DEPs. (**a**,**b**) Detailed subcellular locations (**a**) and statistics histogram (**b**) of the DEPs in RS1-KO retinas at P15 (left) and P56 (right). The number of enriched proteins is labeled after the specific subcellular location. Notably, protein locations do not represent cell type information. (**c**) Composition ratio histogram of DEPs at P15 (left) and P56 (right). RGS, reactome gene sets; BP, biological process.

**Figure 7 cells-11-02150-f007:**
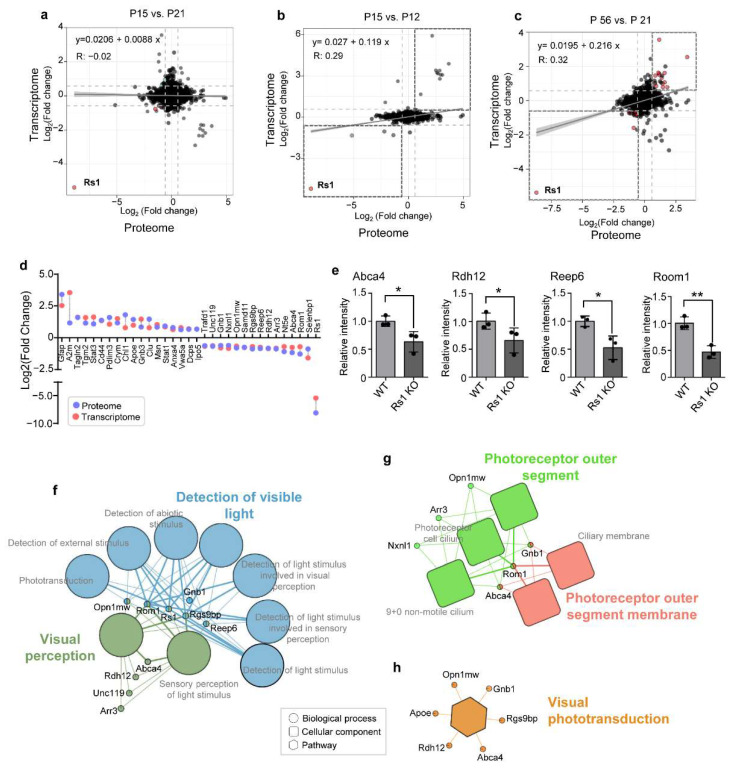
Combined transcriptomic–proteomic analysis of RS1-KO retinas. (**a**–**c**) Similarity correlation analysis of the altered proteome and published transcriptome at different postal days, including P15 vs. P12 (**a**), P15 vs. P21 (**b**), and P56 vs. P15 (**c**). (**d**) Changes in the differentially expressed proteins shared by proteome (P56) and transcriptome (P21) analysis. (**e**) Expression levels of mRNAs of Abca4, Rdh12, Reep6, and Room1. The results are shown as mean ± SD. Differences between groups were assessed by one-way ANOVA. (**f**–**h**) Cytoscape GoClue-based enrichment analysis, including the biological process (**f**), cellular component (**g**), and KEGG pathways (**h**) of the differentially expressed proteins shared by proteome (P56) and transcriptome (P21). Co-enrichment relations of a same color indicates that they are co-enriched in a same subnetwork. Corresponding enrichment results are annotated with the same color. * *p* < 0.05, ** *p* < 0.01.

## Data Availability

The mass spectrometry proteomics data have been deposited to the ProteomeXchange Consortium (http://proteomecentral.proteomexchange.org, accessed on 26 February 2022) via the iProX partner repository with the dataset identifier PXD031901. The data presented in this study are available on request from the corresponding author.

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
