# Peer review of "Retinal Proteomic Alterations and Combined Transcriptomic-Proteomic Analysis in the Early Stages of Progression of a Mouse Model of X-Linked Retinoschisis"

_cells, 2022, doi:10.3390/cells11142150_

Round 1
Reviewer 1 Report
The entire data set, both raw and processed, for the entire proteome must be provided before this can be published. This must to be available on a publicly accessible database, such as the MDPI site. These data must be freely available without any request to the corresponding authors.
Section 4.1 Give a full description of the Rs1 KO mice. How is it generated? Where is the gene alteration? How was this confirmed? Give evidence of full KO of Rs1, with no protein made.
Has this XLRS mouse been described elsewhere in your laboratory publications?
Line 21. “Compared with genotype, age showed a greater impact…“. Please clarify the sentence, as it makes no sense at the moment.
Section 2.7. Lines 262–277. The explanation of the figure is insufficient. The placement on the figure appears to be relative to specific cell types. Is that correct? Have you controlled for secondary effects of degeneration? Please provide your ideas on which of these might be primary and which are simply secondary to general retinal degeneration.
Lines 291 end 296. The genes mentioned here include photoreceptor structural proteins, and you already stated that there is structural pathology and loss of the photoreceptors. So it is self evident that these proteins would be affected. Please clarify and provide some elaboration.
line 371. “This provides further evidence that early intervention may be more effective in remission of the XLRS.“ This is their personal conclusion for which there is not evidence. This must be changed to “we believe this indicates that early intervention may be more effective in remission of the XLRS.“
Reviewer 2 Report
In this study, the authors presented a global proteomic profile of Rs1-KO mice retinas at the onset and early progression stages of XLRS. This study provides insights into the molecular mechanisms underlying the onset and progression of XLRS, which would benefit future XLRS research and treatment development. The manuscript is well-written.
Author Response
Thank you very much for reviewing this manuscript and thank you for your acknowledgment.
Reviewer 3 Report
Retinoschisis (XLRS) is an inherited retinopathy caused by loss of function mutations of the extracellular matrix protein RS1. Many of its features are recapitulated in RS1 knockout mice. Previous transcriptomic studies have compared gene expression between wild-type and RS1-KO retina (ref 6 and Gehrig et al., 2007 – not cited but should be). Here, Jin et al. supplement this work with a mass spec-based proteomic analysis. They profile wild-type and mutant retinas at two ages and analyze the results with an impressive array of computational tools. Based on the analysis, they suggest many potential explanations for the changes they observe.
This is a worthwhile endeavor, in that gene expression and protein abundance are only imperfectly correlated. However, it is difficult to draw decisive conclusions from the data presented. Several results raise doubts. First, the heat maps in Figure 2 seem to show enormous variation among replicates. In other words, there are very few proteins for which all WT samples are red and all RS1 samples are blue – or vice versa. This is quite troubling since it calls into question the significance of the overall trends seen in panels a and b. One sees much less variation among samples in the previous transcriptomic study (e.g., ref. 6, figure 1). Some explanation and analysis of this variability is required Second, many proteins are altered in opposite directions in the mutant at the two time points assayed, P15 and P56 – indeed more than are regulated in the same direction at the two ages (16 vs 15, Figure 4). The authors suggest that “Rs1-KO retinas might exhibit autoregulatory responses” at early ages, but I don’t view this as a plausible explanation for such a large number of proteins. At the very least, one would want to know if these same reversals were observed in the RNAseq study. Third, correlation between mutation-dependent changes detected transcriptomically and proteomically is disturbingly low – r= -0.02, 0.29 and 0.32 for the three comparisons in Figure 6. Some of this undoubtedly reflects the fact that protein abundance is regulated at multiple post-transcriptional levels – but it could also reflect large numbers of inaccuracies in the proteomic data.
In light of these uncertainties some biological validation is required for at least a subset of the changes observed. This would ideally be by immunohistochemistry or western blotting, since protein levels are most relevant, but this could be augmented by qPCR. Lacking that validation, I do not believe that this work will be of significant value to the ophthalmological community.
Specific comments.
Although the paper is generally well-written, some editing for grammar is needed. For example the past tense is used in lines 45, 47, 58, 59, 84 and elsewhere but present tense is correct (for example occurs rather than occurred). On line 52, attributed is used but attributable is correct. And so on.
Line 80: Presumably the histological and ERG results shown in Figure 1 are consistent with previous reports – but this should be stated explicitly, and any differences pointed out.
Lines 103 and 177: The ERG shows a decreased b wave in the mutan. Although this wave reflects bipolar cell activity, it is incorrect to say that only bipolar cells are affected. B wave defects with normal a waves can – and do – also result from decreased neurotransmitter release from photoreceptors.
Line 154 and 200: It seems odd to title the section “differentially expressed gene sets” when the authors have already justified their work by pointing out the important difference between gene expression and protein abundance.
Figure 3c-h. At least this reviewer has no idea whatsoever what the encrichment plots are or how to interpret them. An explanation is required.
Line 187: I would argue that the functional significance of all the RS1-affected pathways remains unknown, not just IFN-1.
Round 2
Reviewer 3 Report
Authors have responded appropriately to criticisms. I do think that their explanations in the rebuttal letter are easier to understand than the ones in the text, and suggest that they enhance the text a bit along the lines of the rebuttal letter.
Author Response
Response: Thanks for your suggestions. Heatmaps of DEPs were added in Figure 2F in the revised manuscript. Evidence figures in explaining the low correlation coefficients in our work was added as supplemental Figure 1. Descriptions in explaining the reliability of our data and the rationale of GSEA was added in the revised manuscript along the lines of the rebuttal letter. (Line 125-130; Line 160-163; Line 343-353)
